# AALawyer: A Generative Retrieval-Augmented Large Language Model System for Legal Reasoning

## ABSTRACT

With the growing potential of large language models (LLMs) in the legal domain, an increasing number of specialized legal models are being developed and introduced. Among them, domain-specific finetuning and retrieval-augmented generation (RAG) methods have received widespread attention. However, there are still some drawbacks such as hallucinated citations and limited explainability. To address these challenges, we construct AALawyer, a generative retrieval-augmented LLM system for legal reasoning in the criminal law domain, and Hallucination Risk-Benchmark, a new benchmark designed for evaluating RAG-based models. Our AALawyer consists of a domain-specific legal LLM named AA-LeLLM and two retrieval modules named AC-RAG and CCs-RAG. Different from both traditional RAG commonly used in legal LLMs and other new RAG, we propose a novel generative RAG, AC-RAG, and construct a CCs-RAG with new criminal cases for retrieval. Experiments demonstrate the professionalism and small hallucination of AALawyer in real-world cases. The model reaches the state-of-the-art level on LawBench classification tasks and scores 88.84% (improve 71.98%) on our target classification task FAP. On the Hallucination Risk-Benchmark, AALawyer outperforms the base model, reducing 37.6% hallucination risk and finally the average score is improved by 31.7%.

## 1 INTRODUCTION

Large Language Models (LLMs) have demonstrated remarkable capabilities in complex natural language understanding and generation tasks across various domains (Wang et al., 2023a; Shi et al., 2024a). In the legal domain, recent studies have shown that LLMs can perform well on legal reasoning, text comprehension (Xiao et al., 2018; Ma et al., 2021), and generation tasks (Zhong et al., 2020). However, legal texts are inherently challenging for general models due to complex terminology, specialized writing styles, and rigorous logical structures. As a result, LLMs often generate inaccurate or hallucinated outputs when applied directly to legal analysis tasks (Huang et al., 2025).

Currently, most specific domain legal LLMs adopt the pipeline "Incremental Pretraining, Finetuning, and Retrieve-Augmented Generation (RAG)" (Huang et al., 2023; Yue et al., 2023; Zhou et al., 2024). Although this strategy improves factual foundation, traditional RAG methods involve a complex workflow that requires fine-tuning, and often retrieves some irrelevant information to have a negative effect on analysis (Yue et al., 2023; Huang et al., 2023). Moreover, many Chinese legal LLMs focus solely on the benchmark score, overlooking the explainability, which is vital for real-world legal applications.

To address these limitations, we propose **AALawyer**, a unified generative RAG-based system that improves legal reasoning through enhanced factual precision and explainability. Our design is inspired by the legal syllogism theory (MacCormick, 1994), which decomposes legal reasoning into three steps: (1) identifying relevant legal norms, (2) analyzing case facts, and (3) reaching a conclusion through legal interpretation. In addition to this theory, real-world judicial decisions often reference precedent court decisions to enhance the persuasiveness of legal reasoning. To reflect this, we also take precedent court decisions into consideration. Combining the legal syllogism theory and real-world judicial practice, AALawyer performs legal reasoning in three stages: retrieving legal ar-

ticles[1], referencing similar precedent cases, and generating conclusions based on case facts. The pipeline is shown in Figure 6 in Appendix. Based on this framework,

Firstly, we define the Hallucination Risk to quantitatively analyze the hallucination. To tackle the issue of hallucination, we propose Article-Content RAG (AC-RAG) to provide legal articles with low hallucination risk, corresponding to the first stage.Our AC-RAG has the following advantages, including (1) **Mitigates the hallucination risk in legal article content**: Our approach ensures that the generated content is always ground-truth, and hallucination is at a low level. (2) **Increases professionalism and explainability in legal analysis**: Traditional legal analysis provides limited reference or explanations, making it difficult for legal practitioners to assess the authenticity of the answers (Dahl et al., 2024; Magesh et al., 2024). In comparison, our method presents relevant legal articles related to the fact case. The evidence makes the analysis more convincing. (3) **Enables effective training**: While the traditional retrieval systems need two models for retrieving and generation, our approach parameterizes retrieval and makes it possible to share the same backbone model for both functions. This reduces the redundant parameters making finetuning simpler.

In addition, we construct a Case-Cases RAG (CCs-RAG) using a large, newly collected dataset to ensure that the generated legal analyses are more comprehensive, professional, and trustworthy, thereby enhancing explainability, which corresponds to the second stage. CCs-RAG has advantages in two aspects. (1) **High retrieval accuracy**: All retrieved $k$ cases closely match the input (shown in the Appendix). We achieve strong performance without finetuning the embedding model. As we say, "expanding the dataset is more effective than tuning the model", which aligns with trends in recent legal LLMs that focus on dataset expansion (Yue et al., 2023; Huang et al., 2023; Cui et al., 2023). (2) **Enhances the explainability and transparency of AALawyer**: Users can view previous case judgments similar to their input. With the final output from LLM, this can provide helpful and reliable references for legal practitioners in making final decisions.

We also finetuned a legal LLM, Authoritative and Accurate Legal LLM (AA-LeLLM), and combined it with the two RAG modules to form our system, AALawyer. And we constructed a 4-dimensional HR-Benchmark for evaluation after RAG. We evaluate the performance on Lawbench, HR-Benchmark and other ablations. As shown in the results section, both our RAG strategies and the model achieve excellent improvement.

Our main contributions include the following.

- We propose a novel generator-retriever-generator pipeline (Generative RAG) and finetune a legal LLM AA-LeLLM oriented on classification and analysis objectives, reducing hallucination risk in legal citation.

- We design a method that retrieves reasoning reference cases to enhance the comprehensiveness and explainability of the generated analysis. To support this, we also collect and organize a new criminal law case dataset and design a method that retrieves reasoning reference cases to enhance the comprehensiveness and explainability of the generated analysis.

- We construct a legal reasoning system grounded in real-world judgments by integrating AA-LeLLM, AC-RAG, and CCs-RAG, resulting in a more professional and hallucination-resistant framework. We also construct a new 4-dimensional benchmark, **HR-Benchmark**, for evaluation after RAG processes.

## 2 PRELIMINARY

### 2.1 RETRIEVAL-AUGMENTED GENERATION MODELS

In the legal domain, the document collection consists of law-related texts such as legal articles, court judgments, and case descriptions. Traditionally, Retrieval-Augmented Generation(RAG) models follow the retriever-and-generator RAG pipeline (Lewis et al., 2020). It first uses a retriever $\eta$ to retrieve some documents $z$ from the given document collection. Then, $z$ is treated as the latent variable that is marginalized via the top-k approximation. The top-k documents occupy the major probability that can approximate the full probability over all documents.

---

[1]In this paper, the term "article" refers specifically to legal provisions or statutes (e.g., Article 234 of the Criminal Law), rather than academic publications, which is in line with conventional legal terminology.

The R&G RAG can approximate the posterior probability of the output sequence $y$. Following Lewis et al. (2020), the posterior probability is

$$p_{\text{R\&G}}(y \mid x) \approx \sum_{z \in \text{top-}k(p_\eta(\cdot \mid x))} p_\eta(z \mid x) \, p_\theta(y \mid x, z), \tag{1}$$

where $x$ denotes the input, $y = (y_1, \ldots, y_N)$ denotes the output sequence generated by the generator model, $p_\eta(z \mid x)$ is the relevance scoring function of the retriever, and $p_\theta(y \mid x, z)$ is the probability that the generator predicts output $y$. $\theta$ is the generator.

Unlike traditional retriever-and-generator pipeline (Lewis et al., 2020) , the generator-and-generator RAG (G&G-RAG) replaces the retriever with a generator $\phi$. Instead of selecting $z$, the generator $\phi$ generates a set of representative intermediate candidates $g$, such as n-gram phrases, entity names, or contextual documents. The posterior probability approximated from G&G RAG can be defined as,

$$p_{\text{G\&G}}(y \mid x) \approx \sum_{g \in \text{top-}k(p_\phi(\cdot \mid x))} p_\phi(g \mid x) \, p_\theta(y \mid x, g). \tag{2}$$

## 2.2 Large Language Model in Law

Existing legal LLMs (Huang et al., 2023; Yue et al., 2023; Zhou et al., 2024) are typically developed following the process of "Incremental Pretraining, Finetuning, and RAG". First, based on an open-source LLM $\theta$, incremental pretraining is conducted on legal general-purpose datasets $D_{\text{law}}$ and other general-domain datasets $D_{\text{general}}$. It is unsupervised training. Then, according to a specific legal task, the model is fine-tuned on the corresponding supervised dataset $D_{\text{law\_task}}$. After finetuning, model $\theta_0$ obtains task-adapted parameters and is used as a generator in the RAG pipeline. Finally, a retrieve-and-generate RAG is employed to construct a prompt used in the final legal analysis task.

## 3 Authoritative and Accurate Lawyer

### 3.1 Article-Content RAG

The existing legal LLMs still suffer from hallucination on the citation of legal articles and their contents (Huang et al., 2025; Yue et al., 2023; Huang et al., 2023). To measure the hallucination, we use the accuracy $\text{Acc}(\theta)$ and the authenticity $\text{Auth}(\theta)$ to measure the hallucinational risk. $\text{Acc}(\theta)$ denotes the accuracy on whether the model cites the correct article number, and $\text{Auth}(\theta)$ measures the correctness of the content in the cited article. Then we define the hallucinational risk as

**Definition 1.** *The Hallucination Risk of the model $\theta$ is*

$$H(\theta) = 1 - Acc(\theta) \cdot Auth(\theta), \quad Acc(\theta), Auth(\theta) \in [0, 1]. \tag{3}$$

Intuitively, a lower dimension output of the generators helps to mitigate hallucination. The output dimension of the second generator in G&G RAG cannot be controlled. So we concentrate on the first generator. To lower the output, we use numeric identifiers to replace the intermediate candidates $g$ in G&G-RAG (Yu et al., 2022), using the constrained generation to achieve the disentangling citation of reasoning. To fit the numeric identifiers, we extend the pipeline of G&G-RAG and propose a new pipeline named AC-RAG.

First, we integrate the article number prediction task into a combined legal dataset for finetuning, enabling the model parameters to learn both downstream legal analysis objectives and article number retrieval capabilities. The model is an Authoritative and Accurate Legal LLM (AA-LeLLM) $\theta_0$.

Then, given the user's input $x$, we format it with a prompt for criminal article number prediction. We convert the generation of $g$ into a classification task by restricting the distribution space of $g$ to a finite set of numeric identifiers. As illustrated in Figure 1a, some standard $g$ are selected and form a sequence. Each $g$ in the sequence can be identified with a unique numeric identifier $n$. Equipped with a classification head, the AA-LeLLM $\theta_0^{\text{cls}}$ predicts a set of numeric identifiers $N$ instead of the

---

**Algorithm 1** AC-RAG

---

**Input:** Legal case input description: $x$; Legal Article Database: $DB_{law} = \{(n_i, c_i)\}_{i=1}^m$; Prompt for article number prediction: $Format_1$; Prompt for final analysis: $Format_2$; Pretrained model parameters: $\theta$; General legal task dataset: $D_{law\_other}$; Specific legal task dataset: $D_{law\_specific}$
**Output:** Final legal analysis $y$
1: $D_{law\_task\_all} = D_{law\_other} \cup D_{law\_specific}$;
2: $\theta_0 = Finetune(\theta, D_{law\_task\_all})$;
3: $x_1 = Format_1(x)$;
4: $\hat{N} \sim p(n \mid x_1; \theta_0)$;
5: **for all** $n_i \in \hat{N}$ **do**
6: $\quad c_i = Retrieve(DB_{law}, n_i)$;
7: $\quad a_i = (n_i, c_i)$;
8: **end for**
9: $x_2 = Format_2(x, \{a_i\}_{i \in \hat{N}})$;
10: $\hat{y} \sim p(y \mid x_2; \theta_0)$;
11: **return** $\hat{y}$;

---

contents. Each predicted number $n_i \in \hat{N}$ can retrieve the corresponding law content $c_i$ from the law article database $DB_{law}$. And we assemble $n_i$ and $c_i$, as $a_i = (n_i, c_i) \in A$.

Finally, the original input $x$ and all retrieved articles $A$ are incorporated into a second prompt for legal analysis. We switch the head of $p_\theta$ into a generation head. Equipped with a generation head, AA-LeLLM $\theta_0^{gen}$ generates the final analysis output $y$. We define the function of AC-RAG as

$$p_{\text{AC-RAG}}(y \mid x) \approx p_{\theta_0}^{cls}(\hat{N} \mid x) \, p_{\theta_0}^{gen}(y \mid x, A), \tag{4}$$

where

$$A = \left\{ (n_i, c_i) \mid n_i \in \hat{N}, \ c_i = DB_{law}[n_i] \right\}. \tag{5}$$

The details of AC-RAG are shown in Algorithm 1.

AC-RAG uses a single model $\theta_0$ to handle both retrieval and output generation effectively. This makes it possible to Generate-Retrieve-Generate pipeline, which differs from previous pipelines, as shown in Figure 1b.

The AC-RAG assigns semantic meaning to the symbols in its parameters, effectively learning a compact representation for retrieval. This helps the dimension of the feature space to stay low. Intuitively, a lower-dimensional feature space is less likely to cause hallucinations. We find that the model's hallucinations are limited to be lower than existing models.

**Lemma 1.** *Denote $\theta$ and $\theta_0$ as the traditional LLM and AC-RAG. Then the hallucination risk of AC-RAG is upper bounded by that of the traditional LLM*

$$H(\theta_0) = 1 - Acc(\theta_0) \leq 1 - Acc(\theta) \cdot Auth(\theta) = H(\theta). \tag{6}$$

*The equality holds only when all generated article content is perfectly accurate and authentic, which cannot be achieved in practice. Proof is shown in the Appendix.*

This Lemma shows that replacing intermediate candidates with numeric identifiers, the hallucination of AC-RAG is guaranteed to be lower than existing G&G RAG models.

## 3.2 CASE-CASES RAG

While AC-RAG can mitigates hallucination risk, it faces more challenges in the legal area. One important challenge is the inherent complexity in legal reasoning. Lawyer LLMs need professional references to assist the judgment in comprehensiveness and explainability. To address this challenge, we design the Case-Cases RAG (CCs-RAG), a retrieval method that can provide relevant precedent court cases to augment the analysis.

CCs-RAG need to be supported by a large number of criminal cases. As there is a lack of appropriate data, we first collected a large-scale dataset $DB_{case}$ of 176k criminal court cases from the publicly

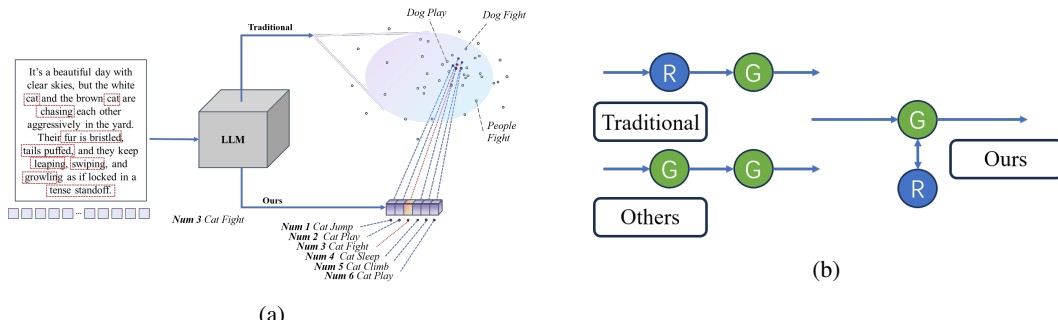

Figure 1: **(a)** A conceptual comparison between the traditional generation-based method and our classification-based method in the output space. **(b)** The pipeline comparison between traditional legal LLM RAG, other generative RAG, and our AC-RAG.

available website. Having these cases we use the embedding model (Xiao et al., 2024) to encode the criminal case dataset into a vector dataset $\{\mathbf{v}_i\}_{i=1}^n$, where each case $c_i \in \mathrm{DB}_{\mathrm{case}}$ corresponds to a vector $\mathbf{v}_i$. Then, the user input $x$ is encoded into $\mathbf{v}_x$ and matched against the case vector dataset using Euclidean distance (Johnson et al., 2019) to retrieve the top-$k$ most relevant cases $\{c_i\}_{i \in \mathcal{C}}$. Next, we construct the prompt $x_1$ by combining $x$, $\{c_i\}_{i \in \mathcal{C}}$, and (if available) $A$ from AC-RAG. Finally, the prompt $x_1$ is fed into the finetuned model $p_\theta$ to generate the final analysis $y$. Algorithm 2 outlines the CCs-RAG retrieval process in the Appendix A.1.3.

### 3.3 AUTHORITATIVE AND ACCURATE LAWYER

In the above methods, AC-RAG mitigates the hallucination risk to provide accuracy and authentication, and CCs-RAG provides the case judgments to make legal reasoning more comprehensive and professional. To combine the above advantages in each method, achieving good performance of legal reasoning, we integrated a three-part framework named AALawyer. The three stages in the AALawyer pipeline are the AC-RAG retrieval, the CCs-RAG retrieval, and final legal analysis by AA-LeLLM, respectively. The whole structure of AA-Lawyer is shown in Figure 2.

**Stage 1 AC-RAG.** The input legal case $e$ is fed into our finetuned model AA-LeLLM to predict the relevant article number(s) $N$. These predicted article numbers are then used to retrieve the corresponding article content $C$ from the AC-Database $DB_{\mathrm{law}}$. Finally, we integrate $e$, $N$, and $C$ into the final input for downstream processing. This stage is shown in the figure in orange.

**Stage 2 CCs-RAG.** We first encode the input case $e$ into a dense vector representation. And then we search our case vector library CCs-Database $DB_{\mathrm{case}}$ to retrieve the top-$k$ most relevant case documents by Euclidean distance. These retrieved documents are then appended to the final input as auxiliary context for enhanced reasoning. This stage is shown in the figure in blue and red.

**Stage 3 Final Analysis Generation.** The final input, comprising the original case $e$, the predicted article number(s) $N$, the article content $C$, and the top-$k$ similar cases, is fed into AA-LeLLM $\theta_0$ to generate the final legal case analysis. In this stage, AA-LeLLM explains the answer with the relevant articles and previous judgments, thereby providing professional and explainable legal reasoning with low hallucination. This stage is shown in the figure in green.

## 4 HALLUCINATION RISK-BENCHMARK

Existing legal benchmark evaluations mostly focus on assessing the model weights alone, lacking a method to evaluate overall system performance. Meanwhile, usual human expert evaluations are too resource-consuming. In order to evaluate the RAG and the effectiveness of our entire system, we leverage other LLMs (Guo et al., 2025) for scoring. Models with larger parameters have stronger logical judgment and analytical professionalism, and can simulate the evaluation of our answers by experts in the legal field, which can be demonstrated in (Zheng et al., 2023; Yue et al., 2023).

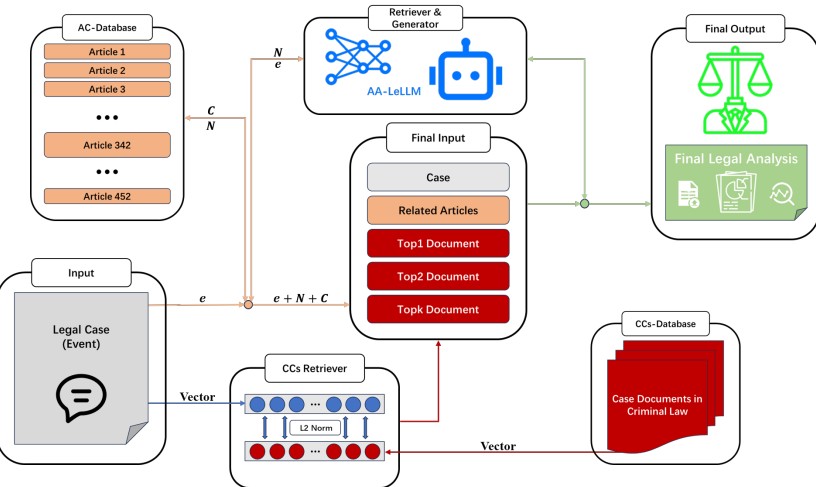

Figure 2: Overall architecture of the AALawyer, consisting of AC-RAG (orange part), CCs-RAG (blue and red part), and the AA-LeLLM generation stage (green part).

We build the Hallucination Risk-Benchmark (HR-Benchmark) to measure the overall performance of the system by focusing on the measurement of modeling hallucination and other aspects. In detail, we evaluate the generated answers from four dimensions they are hallucination score **Hallu**, professionalism **Prof**, informativeness **Info**, and explainability **Expa**. **Hallu↓** is the Hallucination Risk defined in Definition 1. With a lower hallucination score, the model has better performance. **Prof** is the score of the legal professionalism of the analysis. **Info** assesses the richness of information in the analysis. **Expa** refers to the transparency of legal analysis, whether it is supported by exact materials. We also calculate the average score (**Avg.**) to measure the overall performance of the model. The average score is calculated by averaging the value of $(1 - $ **Hallu**$)$ and the other three scores. The scoring range is defined from $0\%$ to $100\%$.

In the evaluation set, we randomly selected 200 cases from the dataset of CAIL2018 (Xiao et al., 2018). To avoid instability in evaluation caused by randomness, we set $temperature$ to 0. When using the API, the evaluations must be conducted at the same period to prevent any potential model updates, ensuring that the evaluation is performed using the same model with identical parameters.

# 5 RELATED WORKS

## 5.1 LEGAL LARGE LANGUAGE MODELS

The early applications of AI in the legal domain focused on information retrieval and extraction (Bommarito II et al., 2021; Ji et al., 2018) , as well as legal prediction and question-resolution tasks (Ma et al., 2021; Ye et al., 2018; Yang et al., 2019; Kien et al., 2020; Zhong et al., 2020), greatly improve the efficiency of legal work.

With the rise of LLMs, their great potential in the legal field has been increasingly recognized. To address domain-specific knowledge limitations and hallucination issues (Huang et al., 2025; Orgad et al., 2024; Rawte et al., 2023; Magesh et al., 2024; Colombo et al., 2024), legal domain LLMs emerged. In Chinese law, these include models based on incremental pretraining and multitask finetuning, such as LawGPT (Zhou et al., 2024), Lawyer LLaMA (Huang et al., 2023), Fuzi.Mingcha (Deng et al., 2023) and LexiLaw (Li et al., 2024); models leveraging finetuning combined with external information retrievals, such as DISC-LawLLM (Yue et al., 2023), ChatLaw (Cui et al., 2023), HanFei (He et al., 2023), and Wisdom-Interrogatory (Wu et al., 2024).

As the field continues to evolve, several standardized benchmarks (Fei et al., 2023; Yue et al., 2023) and legal datasets (Yao et al., 2022; Xiao et al., 2018; Yue et al., 2023; Deng et al., 2023) have been introduced to support this area.

## 5.2 RETRIEVAL-AUGMENTED GENERATION

Retrieval-Augmented Generation (RAG) was first introduced by (Lewis et al., 2020). With the rise of LLMs, RAG has drawn increasing attention in addressing hallucination issues (Yao et al., 2023; Bang et al., 2023). In Gao et al. (2023), RAG is categorized into two types: Naive RAG and Advanced RAG. Naive RAG refers to the traditional Retrieve-and-Generate framework (Chen et al., 2017). In contrast, Advanced RAG achieves better performance by modifying the framework.

Among the Advanced RAG, Modular RAG breaks away from the traditional pipeline, providing some powerful pipelines for reducing hallucinations and enhancing generation quality. Instead, it introduces additional modules such as the Search (Wang et al., 2023b), Memory (Cheng et al., 2023b; Wang et al., 2022), Extra Generation (Yu et al., 2022), Task-Adaptable (Cheng et al., 2023a; Dai et al., 2022), Alignment (Yang et al., 2023; Yu et al., 2023b; Ma et al., 2023), and Validation Module (Yu et al., 2023a). These approaches modify the pipeline in various ways, including Rewrite-Retrieve-Generate (Ma et al., 2023), Generate-Generate (Yu et al., 2022), Recite-Generate (Sun et al., 2022), ITER-Retrieve-Generate (Yang et al., 2024; Shi et al., 2024b), Retrieve-Validate-Generate (Yan et al., 2024). There are some models using a Generate-Generate pipeline. GENRE (De Cao et al., 2020) employs an auto-regressive language model to directly generate entity names as intermediate retrieval targets. SEAL (Bevilacqua et al., 2022) generates n-gram phrases that are likely to appear in relevant documents and uses them as lexical cues for downstream retrieval. GENREAD (Yu et al., 2022) directly generates contextual documents for given questions.

## 6 EXPERIMENT

### 6.1 EXPERIMENT SETUP

We trained the model by incremental pretraining and finetuning, added two RAG processes, and evaluated its performance on Lawbench and HR-Benchmark. The details of datasets and metrics are in the Appendix. All tasks were run on four NVIDIA GeForce RTX 4090 GPUs.

**Incremental Pretraining.** The rank $r$ is set to 8, the scaling factor $\alpha$ is set to 16, and $dropout$ is set to 0. LoRA adaptation is applied across all Transformer layers. The optimizer is AdamW with a learning rate $lr$ of $5 \times 10^{-5}$, and a cosine learning rate scheduler is employed. Gradient accumulation is performed with 8 steps and a batch size of 2 per device, while the cut-off length is set to 2048. The model training is conducted with bfloat16 precision.

**Finetuning.** The per-device batch size is set to 1, while all other settings remain consistent with Incremental Pretraining. We choose DeepSeek-7B as the base model for AA-LeLLM. The model is incrementally pretrained and finetuned using multitask datasets for both legal classification and analysis tasks. The details are shown in the Appendix.

**Evaluation.** In Lawbench, we set the maximum truncation length for inference to 2048 with runs on 500 samples. In HR-Benchmark, $temperature$ is 0. Evaluation model is $deepseek\text{-}chat$ and runs on 200 randomly selected samples.

### 6.2 MAIN RESULTS

As shown in Table 1, the Criminal Article Classification task (FAP), achieves a significant improvement of 71.98% over the base DeepSeek-7B model, reaching the best classification performance among all compared models. Similarly, the Criminal Case Classification task (CP) also improves by 29.04%. For full-domain legal classification tasks, Marital Disputes Identification (MDI) improves by 6.60%, Issue Topic Identification (ITI) by 3.20%, and Event Detection (ED) by 23.01%. Although Dispute Focus Identification (DFI) drops by 7.10%, this is expected since it belongs to the full legal domain and is not aligned with our training goal, which is specifically focused on criminal law. Even so, the performance remains within a reasonable range.

In Table 2a, compared with our base model DeepSeek-7B, our AALawyer reduces hallucination risk by 37.6%, improves professionalism by 13.4%, informativeness by 41.9%, and explainability by 43.4%, and achieves an overall average score increase by 34.1%. Furthermore, compared with our AA-LeLLM without RAG, our AALawyer reduces the 12.2% hallucination risk, improves 10.3%

Table 1: Scores of models with comparable parameter sizes on LawBench classification tasks.

| Model | Criminal | | Full Domain | | | | Average |
|---|---|---|---|---|---|---|---|
| | FAP | CP | DFI | MDI | ITI | ED | |
| DeepSeek-7B (Guo et al., 2025) | 16.86 | 28.89 | 27.20 | 39.69 | 35.80 | 58.35 | 34.47 |
| LawGPT-beta1.1-7B (Zhou et al., 2024) | 0.15 | 15.68 | 4.95 | 6.85 | 2.40 | 14.94 | 7.50 |
| LexiLaw-6B (Li et al., 2024) | 13.15 | 39.99 | 3.30 | 15.60 | 22.80 | 15.30 | 18.36 |
| HanFei-7B (He et al., 2023) | 2.64 | 30.96 | 6.39 | 30.44 | 30.20 | 14.73 | 19.23 |
| Wisdom-Interrogatory-7B (Wu et al., 2024) | 32.84 | 35.09 | 7.84 | 36.72 | 21.00 | 15.98 | 24.91 |
| Fuzi-Mingcha-7B (Deng et al., 2023) | 25.19 | 55.93 | 19.59 | 28.46 | 18.60 | 16.90 | 27.45 |
| Qwen3-8B (Team, 2025) | 73.21 | 51.88 | 46.00 | 54.40 | 38.80 | 65.86 | 55.02 |
| Internlm3-8B (Cai et al., 2024) | 82.92 | 55.02 | 37.20 | 52.29 | 41.20 | 66.38 | 55.83 |
| AA-LeLLM-7B(Ours) | 88.84 ± 0.34 | 57.93 ± 0.60 | 20.10 ± 0.50 | 46.29 ± 1.19 | 39.00 ± 1.52 | 81.36 ± 0.07 | 55.59 |

professionalism, $57.1\%$ informativeness, and $47.3\%$ explainability, and achieves an overall average score increase $31.7\%$.

Overall, AA-LeLLM demonstrates excellent performance on our target classification task, FAP, enabling AC-RAG to resolve hallucination issues while maintaining high accuracy, resulting in effective legal reasoning. Moreover, AA-LeLLM also shows competitive performance on other non-target classification tasks in various datasets, achieving an average score improvement of $21.12\%$ compared to the base model. This highlights its robustness and effectiveness as a legal classifier. It serves as a qualified module for the retriever component of AC-RAG in our AALawyer system. In the system, AC-RAG and CCs-RAG both show strong performance on HR-Benchmark, AC-RAG mainly improves **Hallu↓**, **Prof** and **Expa**, while CCs-RAG mainly improves **Info** and **Expa** of legal reasoning. The low **Hallu↓** of AC-RAG also proves Lemma 1.

### 6.3 ABLATION STUDIES

#### 6.3.1 CALCULATION OF HALLUCINATION RISK

To further prove our Lemma 1, we selected 150 cases from the dataset of CAIL2018 (Xiao et al., 2018) and calculate the hallucination risk $\mathrm{H}(\theta)$ on the latest criminal law as

$$\mathrm{H}(\theta) = \frac{1}{N} \sum_{i=1}^{N} \left(1 - \mathrm{Acc}_i(\theta) \cdot \mathrm{Auth}_i(\theta)\right) \tag{7}$$

The results are shown in Table 2b and the detailed metrics are shown in the Appendix A.1.2. We calculate the average score of $\mathrm{Acc}_i(\theta)$ and $\mathrm{Auth}_i(\theta)$. The Hallu↓ is calculated according to Formula 7, rather than directly from $Avg.Acc$ and $Avg.Auth$. The result shows that our AC-RAG has a score of $0.93$ in $Avg.Acc$, indicating that it effectively mitigates the $\mathrm{Auth}(\theta)$ component of the hallucination risk. And AC-RAG reduces hallucination risk by $59\%$ compared with base model, which shows an excellent performance and achieves the goal of our AC-RAG. Furthermore, these metric is grounded in a mathematically defined hallucination risk formula, enabling a more objective and reliable assessment of hallucination. The results are corresponding to the result tables in Main Result part, which also validate the reliability of our proposed HR-Benchmark.

#### 6.3.2 COMPARED WITH OTHER TYPES OF RETRIEVAL

A single case may correspond to multiple legal articles due to the inherent complexity of legal cases. Traditional multi-label classification methods often rely on fixed thresholds $\tau$ as $\hat{A} = \{j \mid \hat{y}_j \geq \tau\}$ or by selecting the top-$k$ scoring labels as $\hat{A}_{\text{top-}k} = \arg \text{top-}k_{j=1,\ldots,K} \hat{y}_j$. Both have problems. As shown in Table 3, when $\hat{y}_n \approx \tau$, it is unclear whether that label should be included, and the optimal value of $\tau$ can vary by case. Alternatively, top-$k$ strategies avoid threshold tuning but might include low-confidence labels, risking irrelevant results. In contrast, AC-RAG trains the model to learn the mapping between case textual features and legal article numbers, allowing it to decide how many labels to output without relying on rigid decision boundaries, thus avoiding the pitfalls of traditional

Table 2: **(a)** Comparison of our AA-LeLLM against DeepSeek-7B (DS-7B) on HR-Benchmark. **(b)** Hallucination risk calculation by metrics.

(a)

| Baseline | Method | Hallu↓ | Prof | Info | Expa | Avg. |
|---|---|---|---|---|---|---|
| DS-7B | Vanilla | 82.4 | 58.8 | 40.0 | 44.6 | 40.2 |
| | AC-RAG | 85.4 | 46.4 | 39.4 | 44.2 | 36.2 |
| | CCs-RAG | 63.0 | 66.6 | 81.6 | 78.8 | 66.0 |
| | AALawyer | 81.4 | 55.0 | 74.0 | 70.8 | 54.6 |
| AA-LeLLM | Vanilla | 57.0 | 61.9 | 24.8 | 40.7 | 42.6 |
| | AC-RAG | **39.6** | 70.6 | 37.9 | 67.8 | 59.1 |
| | CCs-RAG | 53.9 | 60.9 | 74.7 | 79.5 | 65.3 |
| | AALawyer | 44.8 ± 1.8 | **72.2** ± 0.2 | **81.9** ± 0.2 | **88.0** ± 0.2 | **74.3** |

(b)

| Baseline | Method | Avg.Acc | Avg.Auth | H(θ)↓ |
|---|---|---|---|---|
| DS-7B | Vanilla | 0.29 | 0.44 | 0.88 |
| Qwen3-8B | Vanilla | 0.59 | 0.70 | 0.58 |
| | AC-RAG | 0.59 | 0.94 | 0.44 |
| Internlm3-8B | Vanilla | 0.71 | 0.69 | 0.51 |
| | AC-RAG | 0.71 | 0.95 | 0.33 |
| AA-LeLLM | Vanilla | 0.76 | 0.49 | 0.63 |
| | AC-RAG | 0.76 | 0.93 | **0.29** |

Table 3: Performance comparison of different retrieval methods on the FAP task. Results are shown for different parameter: top-k ($k$) and thresholds ($\tau$). The dagger (†) indicates the optimal value for the parameter. The optimal parameter selections are shown in the Appendix.

| Category | Method | Parameter | F1-Score | Precision | Recall |
|---|---|---|---|---|---|
| Dense Retrieval | BGE-v1.5 (Xiao et al., 2024) | $k^{\dagger} = 1$ | 45.80 | 45.80 | 45.80 |
| | | $\tau^{\dagger} = 0.78$ | 35.83 | 31.47 | 41.60 |
| | BGE-m3 (Chen et al., 2024) | $k^{\dagger} = 1$ | 50.60 | 50.60 | 50.60 |
| | | $k = 2$ | 41.73 | 31.30 | 62.60 |
| | | $k = 3$ | 33.80 | 22.53 | 67.60 |
| | | $\tau^{\dagger} = 0.63$ | 36.53 | 43.29 | 31.6 |
| Sparse Retrieval | BM25 | $k^{\dagger} = 1$ | 33.40 | 33.40 | 33.40 |
| | | $k = 2$ | 27.60 | 20.70 | 41.40 |
| | | $k = 3$ | 23.40 | 15.60 | 46.80 |
| Generative Retrieval | AC-RAG (Qwen3-8B) | - | 73.20 | 67.70 | 74.83 |
| | AC-RAG (Internlm3-8B) | - | 82.92 | 43.81 | 78.39 |
| | AC-RAG (AALawyer) | - | **88.94** | **89.67** | **84.26** |

methods. This adaptive threshold enables more reliable predictions across diverse inputs, as

$$\hat{N} = \{n \mid p_{\theta}^{\text{cls}}(x)[n] \geq \tau_{\theta}^{*}(x)\}. \tag{8}$$

### 6.3.3 RAG WITHOUT FINETUNING

Table 2a shows the non-finetuned base model DeepSeek-7B performance with each RAG. The results are consistent with our prior theory. AC-RAG performs poorly on DeepSeek-7B because it need to be used combined by special finetuning process, corresponding with Algorithm 1. And it shows a good performance with CCs-RAG, because CCs-RAG is supported by a huge and professional criminal law dataset, making it highly effective for criminal law tasks regardless of model weights. This demonstrates that it is a robust and adaptable RAG method in the criminal law domain.

More other ablations are shown in the Appendix A.2.

## 7 CONCLUSION

In this paper, we introduced AALawyer, a legal analysis assistant system focused on criminal law. Inspired by legal syllogism and real-world judicial processes, the system is designed to reduce hallucinations in legal citation and enhance legal reasoning. Within this system, we proposed a novel Generative RAG pipeline (AC-RAG) and case retrieval module (CCs-RAG), in which we collected a new dataset of 176k criminal case judgments. Additionally, we finetuned a legal LLM, AA-LeLLM, oriented toward classification and analysis tasks in law. Furthermore, we constructed HR-Benchmark, a 4-dimensional benchmark, to evaluate the content after RAG. Experimental results show that AA-LeLLM, AC-RAG, and CCs-RAG all achieve strong performance in legal reasoning.

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

# A APPENDIX

## A.1 THEORETICAL ANALYSIS

### A.1.1 PROOF OF LEMMA 1

By definition, the hallucination risk for the traditional model $\theta$ is

$$H(\theta) = 1 - \text{Acc}(\theta) \cdot \text{Auth}(\theta).$$

$$\text{Acc}(\theta) = \frac{2 \cdot |f_\theta(x) \cap y|}{|f_\theta(x)| + |y|}, \quad f_\theta(x) \subseteq A, \quad y \subseteq A$$

$$\text{Auth}(\theta) = \frac{(1 + \beta^2) \cdot \text{LCS}(g_\theta(x), c_y)}{\text{len}(g_\theta(x)) + \beta^2 \cdot \text{len}(c_y)}$$

So our objective to reduce hallucination risk is

$$\min_\theta \quad H(\theta) = 1 - \text{Acc}(\theta) \cdot \text{Auth}(\theta).$$

Both $\text{Acc}(\theta)$ and $\text{Auth}(\theta)$ must be increased to reduce $H(\theta)$.

The overall AC-RAG prediction process can be expressed as

$$p_{\text{AC-RAG}}(y \mid x) \approx p_{\theta_0}^{\text{cls}}(\hat{N} \mid x)\, p_{\theta_0}^{\text{gen}}(y \mid x, A),$$

where

$$A = \left\{ (n_i, c_i) \mid n_i \in \hat{N}, \ c_i = DB_{\text{law}}[n_i] \right\}.$$

For our AC-RAG model $\theta_0$, because the article content $c_i$ is retrieved from a ground-truth database rather than generated, we can assume the authenticity is perfect, $\text{Auth}(\theta_0) = 1$. The risk for our model thus simplifies as

$$H(\theta_0) = 1 - \text{Acc}(\theta_0).$$

To prove the lemma, $H(\theta_0) \leq H(\theta)$, we need to show that:

$$1 - \text{Acc}(\theta_0) \leq 1 - \text{Acc}(\theta) \cdot \text{Auth}(\theta)$$

$$\text{Acc}(\theta_0) \geq \text{Acc}(\theta) \cdot \text{Auth}(\theta). \tag{9}$$

We now justify this premise by analyzing the optimization objectives.

Therefore, the original 2-dimensional optimization is reduced to a single-dimensional objective, shown as

$$\min_{\text{Acc, Auth} \in [0,1]} H(\theta) = 1 - \text{Acc} \cdot \text{Auth} \implies \min_{\text{Acc} \in [0,1]} H(\theta_0) = 1 - \text{Acc}.$$

So the optimized gradient

$$\nabla_\theta H(\theta) = \frac{\partial H(\theta)}{\partial \text{Acc}(\theta)} \cdot \frac{\partial \text{Acc}(\theta)}{\partial \theta} + \frac{\partial H(\theta)}{\partial \text{Auth}(\theta)} \cdot \frac{\partial \text{Auth}(\theta)}{\partial \theta}$$

$$= - \left( \text{Auth}(\theta) \cdot \frac{\partial \text{Acc}(\theta)}{\partial \theta} + \text{Acc}(\theta) \cdot \frac{\partial \text{Auth}(\theta)}{\partial \theta} \right)$$

becomes

$$H(\theta_0) = 1 - \text{Acc}(\theta_0) \quad \Rightarrow \quad \nabla_{\theta_0} H(\theta_0) = -\frac{\partial \text{Acc}(\theta_0)}{\partial \theta_0}.$$

The training of the traditional model $\theta$ involves a two-dimensional optimization, where the gradient is a composite signal from two potentially conflicting objectives to improve accuracy and authenticity simultaneously.

In contrast, the training of our model $\theta_0$ is a simpler, single-dimensional problem focused solely on maximizing $\text{Acc}(\theta_0)$. The gradient is a more direct signal. Given that the optimization of $\theta_0$ is more focused and stable, it is reasonable to conclude that it can more effectively achieve a final accuracy $\text{Acc}(\theta_0)$ that surpasses the $\text{Acc}(\theta) \cdot \text{Auth}(\theta)$ which has the more complex training process. Therefore, the premise Equation 9 is justified, which completes the proof.

### A.1.2 THE METRICS OF HALLUCINATION RISK CALCULATION

We use the metric

$$\text{Auth}_i(\theta) = \frac{(1 + \beta^2) \cdot \text{LCS}(g_\theta(x_i), c_{y_i})}{\text{len}(g_\theta(x_i)) + \beta^2 \cdot \text{len}(c_{y_i})}, \tag{10}$$

where we set $\beta = 0$, and

$$\text{Acc}_i(\theta) = \frac{2 \cdot |f_\theta(x_i) \cap y_i|}{|f_\theta(x_i)| + |y_i|}, \quad f_\theta(x_i), y_i \subseteq A \tag{11}$$

in hallucination risk $H(\theta)$ calculation.

### A.1.3 ALGORITHM OF CCS-RAG

Shown in Algorithm 2.

---

**Algorithm 2** CCs-RAG

---

**Input:** User input description: $x$; Legal case database: $\text{DB}_{\text{case}} = \{c_1, c_2, \ldots, c_n\}$; All legal case vectors: $\{\mathbf{v}_i\}_{i=1}^n$; Encoder: $\text{Enc}(\cdot)$; Prompt formatter: $\text{Format}$; Generator: $p_\theta(y \mid \cdot)$; Top-$k$: $k$; Maximum token length: $T_{\text{max}} = 2048$; *(Optional)* Relevant legal article set: $A = \{(n_i, c_i)\}_{i=1}^m$

**Output:** Final legal analysis $\hat{y}$
1: $\mathbf{v}_x = \text{Enc}(x)$;
2: $\mathcal{C} = \text{k-argmin}_{i=1}^n \|\mathbf{v}_x - \mathbf{v}_i\|_2$;
3: **for all** $i \in \mathcal{C}$ **do**
4: $\quad c_i = \text{Retrieve}(\text{DB}_{\text{case}}, i)$;
5: **end for**
6: $x_1 = \text{Format}(x, \{c_i\}_{i \in \mathcal{C}}, A)$;
7: $\hat{y} \sim p_\theta(y \mid x_1), \quad \text{s.t. } \text{TokenLen}(x_1) \leq T_{\text{max}}$;
8: **return** $\hat{y}$;

---

## A.2 ADDITIONAL EXPERIMENTS

### A.2.1 SELECTION OF OPTIMAL THRESHOLD

As shown in Figure 3b, we evaluate the F1-score for each threshold of the RAG models, and finally present the optimal values in the main evaluation. The optimal thresholds were tested to be $0.78$ for BGE-v1.5 and $0.63$ for BGE-m3.

### A.2.2 CCS-RAG SCALE

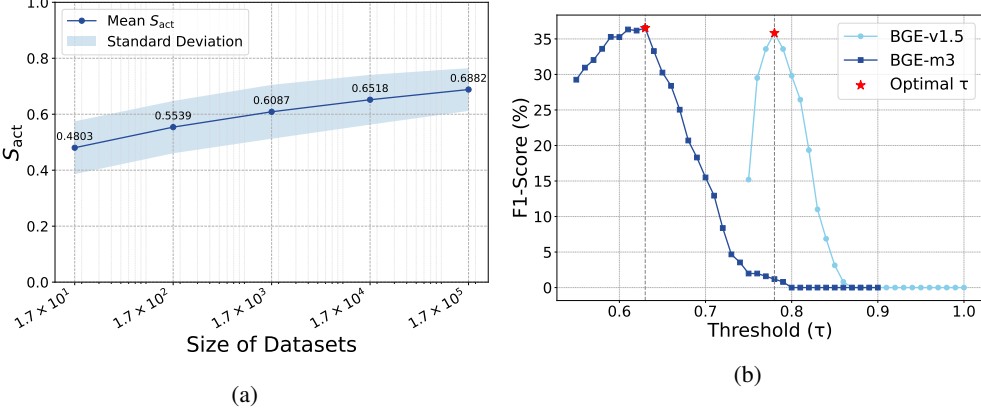

(a)

(b)

Figure 3: (a) The effectiveness of our large-scale datasets. (b) The optimal thresholds of RAG models.

To evaluate the retrieval effectiveness of our large-scale legal case dataset within the CCs-RAG framework, we designed an ablation study to measure performance across varying data scales. We find that original score of vectors similarity $S_{\text{ori}}$ will produce high scores even for irrelevant queries, so we test the base noise $\epsilon$ and calculate a Actual Score $S_{\text{act}}$ as

$$S_{\text{act}} = \frac{S_{\text{ori}} - \epsilon}{1 - \epsilon} \tag{12}$$

To determine our base noise $\epsilon$, we measured the system performance using 10 meaningless text strings. We calculated the average top-10 similarity score for each string over 100 independent runs. This test yielded a mean score of $0.6652 \pm 0.0196$. To ensure a effective (positive) result in our subsequent calculations, we defined $\epsilon$ as the lower bound of this noise range: $\epsilon = 0.6456$.

Then we measured $S_{\text{act}}$ by running the experiment 20 times for 5 law cases randomly selected from the China Court Website. This evaluation was performed on subsets of our entire dataset, randomly

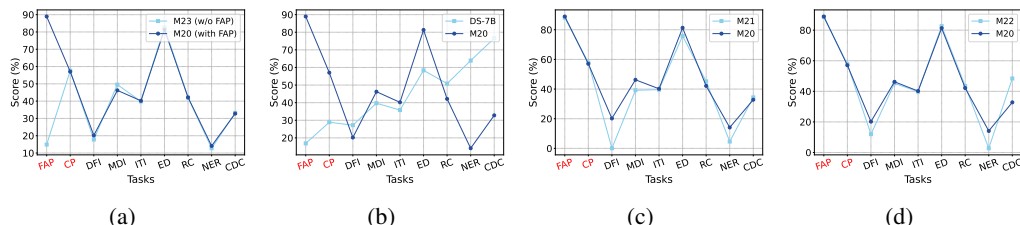

(a)                    (b)                    (c)                    (d)

Figure 4: Comparison between different versions of AA-LeLLM: (a) Training without FAP, (b) $M_{20}$ vs. DS-7B, (c) $M_{20}$ vs. $M_{21}$, (d) $M_{20}$ vs. $M_{22}$. $M_{20}$ is our final version.

sampled at scales of 17, 170, 1,700, 17,000, and 170,000 documents at each running time, with the average top-10 $S_{\text{act}}$ being calculated. The results in Figure 3a shows that a larger dataset leads to the retrieval of cases that are more similar to the query, which demonstrate the effectiveness of our large-scale datasets in CCs-RAG.

### A.2.3 TRAINING WITHOUT AC-RAG

For the completeness of the experiment, we excluded the target task FAP from the finetuning, resulting in $M_{23}$, as shown in Figure 4a. We can observe that the results of other tasks only have minor fluctuations. This further demonstrates the effectiveness of our proposed AC-RAG training approach, which does not negatively impact the performance of other finetuning tasks. The primary focus of this work is to validate the effectiveness of AC-RAG. So we are not focus on task-specific data augmentation or finetune for other task objectives. If further work intends to improve performance on other non-oriented tasks, expanding the finetuning data or methods would be a viable strategy. And integrating our AC-RAG retrieval module into the training pipeline does not interfere with the performance of other task objectives.

We trained multiple versions of AA-LeLLM and finally selected $M_{20}$ as our final model. The comparisons are shown in Figure 4. The training differences, detailed results, and further analyses are in the Appendix A.2.5.

### A.2.4 TRAINING-FREE DEPLOYMENT WITH SOTA LLMS

We have observed that as large models evolve, SOTA models are approaching the performance of our fine-tuned AA-LeLLM on the FAP task (around 80%), as shown in Table 1. This indicates the possibility of a training-free AC-RAG, where general models already has the necessary legal prediction and analysis capabilities without specific fine-tuning. To assess this, we conducted an ablation study on SOTA models using the latest version of the criminal law. We find an advantage of our AC-RAG method that when laws are modified, we only need to update our legal article retrieval database. Since the crime corresponding to an article number generally remains unchanged during modified, this approach avoids the hard process of re-annotating data when it becomes a new version of law. This is proved by the result in Table 4 that our baselines (DS-7B and AA-LeLLM) maintained trends consistent with our main tables with old version of law contents.

The recent SOTA level model Qwen3-8B showed a different trend: AC-RAG improved overall performance but unexpectedly increased the Hallu↓. We check the result and find that this stems not from a failure in legal reasoning, but from the model's inability to follow the unseen prompt format for the information-sparse numeric prediction task. This suggests that finetuning remains necessary for now, and a better-designed prompt could be key for future training-free deployment. Meanwhile, our Table 2b shows that SOTA models are becoming increasingly accurate at the prediction task, reaching a high level, suggesting that as foundation models evolve, they may acquire sufficient latent legal knowledge to make AC-RAG a truly training-free method.

### A.2.5 DATASETS SELECTION OF TRAINING PROCESS

We experimented with three different training strategies, resulting in $M_{20}$, $M_{21}$, and $M_{22}$. After careful consideration, we selected $M_{20}$ as our final AA-LeLLM into the AALaywer. The training

Table 4: Comparison of models on HR-Benchmark on the latest criminal law (different setting with main part table).

| Baseline | Method | Hallu↓ | Prof | Info | Expa | Avg. |
|---|---|---|---|---|---|---|
| DS-7B (non-finetuned) | Vanilla | $77.22 \pm 0.63$ | $56.13 \pm 0.46$ | $38.73 \pm 0.12$ | $45.63 \pm 0.35$ | $40.82 \pm 0.33$ |
| | AC-RAG | $79.76 \pm 0.41$ | $45.93 \pm 0.35$ | $37.27 \pm 0.40$ | $43.57 \pm 0.45$ | $36.75 \pm 0.36$ |
| | AALawyer | $74.30 \pm 0.18$ | $53.12 \pm 0.07$ | $72.72 \pm 0.27$ | $66.31 \pm 0.34$ | $54.46 \pm 0.17$ |
| Qwen3-8B (non-finetuned) | Vanilla | $31.21 \pm 0.68$ | $72.00 \pm 0.36$ | $35.60 \pm 0.10$ | $57.50 \pm 0.10$ | $58.47 \pm 0.28$ |
| | AC-RAG | $36.61 \pm 0.50$ | $77.23 \pm 0.05$ | $48.63 \pm 0.23$ | $75.63 \pm 0.25$ | $66.22 \pm 0.20$ |
| | AALawyer | $29.74 \pm 0.14$ | $92.87 \pm 0.32$ | $97.77 \pm 0.05$ | $96.93 \pm 0.32$ | $89.45 \pm 0.03$ |
| AA-LeLLM (finetuned) | Vanilla | $46.77 \pm 0.43$ | $64.07 \pm 0.12$ | $28.40 \pm 0.10$ | $48.53 \pm 0.46$ | $48.56 \pm 0.21$ |
| | AC-RAG | $31.68 \pm 0.06$ | $74.97 \pm 0.21$ | $42.00 \pm 0.10$ | $73.63 \pm 0.15$ | $64.73 \pm 0.07$ |
| | AALawyer | $36.40 \pm 0.44$ | $70.73 \pm 0.15$ | $78.70 \pm 0.17$ | $84.70 \pm 0.40$ | $74.43 \pm 0.05$ |

Table 5: Datasets of training: We use a number to represent the source. (**1**) fuzi.mingcha (Deng et al., 2023) (**2**) CAIL2018 (Xiao et al., 2018) (**3**) DISC (Yue et al., 2023) (**4**) law-lib.

| Process | Source | Type | Num | Size | $M_{20}$ | $M_{21}$ | $M_{22}$ | $M_{23}$ |
|---|---|---|---|---|---|---|---|---|
| IPT | (**1**) | Full Domain Chinese Legal Articles | 57k | 21MB | ✓ | ✓ | ✓ | ✓ |
| | (**1**) | Articles of the Criminal Law | 452 | 266kB | ✓ | ✓ | ✓ | ✓ |
| | (**1**) | Legal Case Documents | 1k | 16MB | ✓ | ✓ | ✓ | ✓ |
| | (**4**) | Case Documents in Criminal Law | 172k | 1052MB | | ✓ | | |
| SFT | (**2**) | Case - Charge(s) | 155k | 248MB | ✓ | ✓ | ✓ | ✓ |
| | (**2**) | Case - Article Number(s) | 155k | 251MB | ✓ | ✓ | ✓ | |
| | (**2**) | Case - Criminal | 155k | 250MB | ✓ | ✓ | ✓ | ✓ |
| | (**2**) | Case - Fine | 155k | 244MB | ✓ | ✓ | | ✓ |
| | (**2**) | Case - Sentencing | 155k | 252MB | ✓ | ✓ | | ✓ |
| | (**3**) | Criminal Law QA Events Analyses | 13k | 43MB | ✓ | ✓ | ✓ | ✓ |
| | (**3**) | Full Legal Domain QA Events Analyses | 19k | 69MB | ✓ | ✓ | ✓ | ✓ |
| | (**3**) | Full Legal Domain QA Tasks | 205k | 361MB | ✓ | ✓ | ✓ | ✓ |
| AC-RAG | (**1**) | Criminal Article Number - Content | 452 | 164kB | ✓ | ✓ | ✓ | ✓ |
| CCs-RAG | (**4**) | Vector of Case in Criminal Law | 172k | 676MB | ✓ | ✓ | ✓ | ✓ |

datasets used for each model are shown in Table 5. The comparison results are displayed in Figure 4, where we compare the performance of all Group 1 classification tasks and those tasks in Group 2.

**Change in Incremental Pretraining.** We explored different incremental pretraining strategies to generate $M_1$. One strategy included only all law articles, criminal law articles and some case documents, resulting in $M_{10}$, while the other added extra case documents from criminal law to the training data, generating $M_{11}$. $M_{10}$ was fine-tuned to produce $M_{20}$, and $M_{11}$ was fine-tuned to produce $M_{21}$. The results showed that the model $M_{21}$, which was trained with more data for incremental pretraining, performed worse. This is because our new data consisted of pure criminal law cases, which affected the general legal classification tasks. Additionally, our focus tasks in criminal law, FAP and CP, did not show significant improvement on $M_{21}$. Therefore, we chose $M_{10}$ for the subsequent SFT stage.

**Change in SFT.** In the SFT task, we excluded the amount prediction and sentence prediction tasks that influenced CDC (Criminal Damages Calculation) and then fine-tuned $M_{10}$ to produce $M_{22}$. The results showed that CDC performance improved, confirming our later reasoning that these two tasks indeed affected the model's mathematical operations and logic. However, our AALawyer does not require mathematical logic computation, and the focus identification ability of the DFI (Dispute Focus Identification) model in $M_{22}$ declined, which is more important for our model. Therefore, we decided to continue including the amount prediction and sentence prediction training data.

### A.2.6 SINGLE TASK TRAINING

When we use single-task training, we find that only the training task can work well, as shown in Table 6. So we choose the multi-task training (Caruana, 1997; Yue et al., 2023), combining multiple tasks' training data, as shown in line $M_{20}$ of Table 5, to train our model. We can see that the performance of $M_2$-multi works well in all tasks, maintain the good performance from DeepSeek-

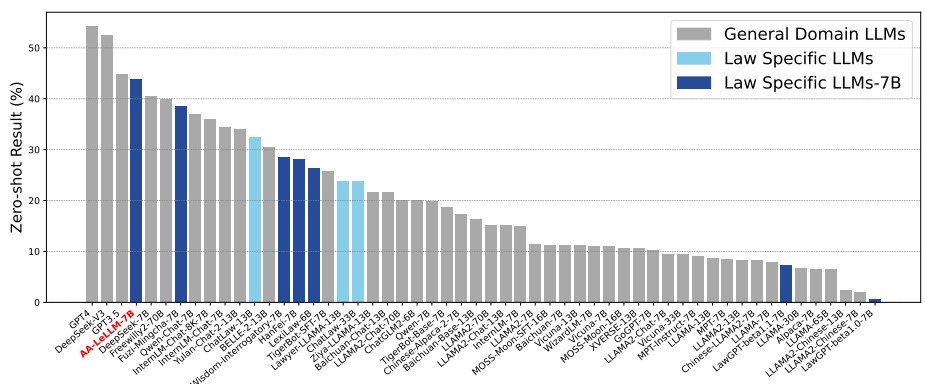

Figure 5: Full domain evaluation on LawBench.

7B and outperforms in our classification task. However, comparing with single-task training, the performance shows slight degradation in CP, but the impact is small, and other tasks perform excellently. For our multi-task training requirements in AALawyer and training efficiency, this approach is clearly better, with minimal impact on the valuable pretrained weights of DeepSeek-7B.

### A.2.7 EVALUATION ON FULL DOMAIN LEGAL TASKS

We further calculate the average score of 11 metrics in LawBench (Fei et al., 2023), comparing with the base model and other models. As shown in the Figure 5, despite being tested on general-domain tasks, our model still maintained a strong level of performance. Since we focus on the classification and analysis of criminal law, and the training data primarily consists of single-label classification and legal analysis tasks, performance on some unseen tasks in some Full-domain legal evaluation is suboptimal. However, the model still maintains a good overall performance. Among all models with comparable parameter sizes, it achieved the SOTA level.

Our performance on some unseen tasks in the full-domain legal evaluation is suboptimal. However, the following analysis indicates that these tasks do not impact the primary objectives of AALawyer. The results are shown in Table 7.

Regarding RC (Reading Comprehension), the reference answers are short, but our model's answers include more analysis (long), leading to a lower score. However, the answers are mostly correct, just with some differences in similarity to the reference examples (the examples are short, and ours are longer), but longer answers are more suitable for our analysis task. The model's performance is not good in NER (Named-Entity Recognition) because our SFT training data only includes tasks related to recognizing criminals, and it doesn't involve other entity recognition tasks, which means it can only recognize criminals. However, as our main tasks are classification and analysis, primarily focusing on identifying the criminal and the relevant legal article numbers, this metric does not have a significant effect. OS (Opinion Summarization) metric is effective for case analysis to supports legal analysis tasks. The model's performance is well. CDC (Criminal Damages Calculation) evaluate the model's numbers extraction and mathematical ability. Due to the presence of tasks related to predicting criminal financial amounts in the SFT training data, this may affected normal calculations. After testing with standard mathematical summation tasks, the results were correct. So we deem that the math level maintains a enough level. And the target tasks do not involve financial amount calculation, so this metric has little impact. Co (Consultation) questions are general law issues, and since our model specializes in criminal law, it can't answer the question in other legal domains accurately, but the results are comparable to other legal domain-specific models.

### A.2.8 SHOWCASES

We randomly selected a case in the Chinese Court Website as input, and the showcase result is shown in the Table 8. In this real-world case analysis, our AALawyer accurately find the relevant legal articles, generates professional legal reasoning, and retrieves highly similar cases. This demonstrates

its high accuracy in legal citation and strong explainability. And even when increasing CCs-RAG cases $k$ to 5 or varying the input $x$, the retrieved cases are still closely match to the input.

## A.3 ADDITIONAL DETAILS

### A.3.1 BENCHMARK AND DATASETS

**Benchmarks.** We use 11 metrics in Lawbench, and a 4-dimension HR-Benchmark.

LawBench is a legal benchmark designed based on Bloom's cognitive model, covering Knowledge Memorization, Knowledge Understanding, and Knowledge Applying. We chose LawBench as the benchmark for model ability testing. The evaluation results are in the main paper under the Main Results section. Before evaluation, we analyzed the principles and objectives of the related metrics, removed metrics that were ineffective for all models and excluded all multiple-choice questions. In the end, we leave 11 relevant metrics and created an automated evaluation script to assess each model. The selected metrics are divided into two groups:

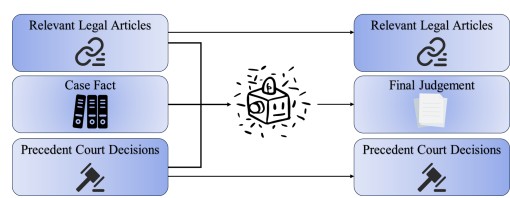

Figure 6: The system concept based on the legal syllogism and actual legal judgment.

The 1st group is based on classification-related tasks in LawBench, which are consistent with our project model's training goals, as shown in Main Result. It includes 2 criminal law domain MLC (Multi-label Classification) tasks: FAP and CP, 3 full legal domain MLC tasks: DFI, MDI, and ED, and 1 SLC (Single-label Classification) task: ITI. The evaluation details and metrics for each task are as follows:

**FAP** (Fact-based Article Prediction) predicts the relevant criminal law article numbers for a given event. **CP** (Charge Prediction) predicts the criminal charge related to a given event. **DFI** (Dispute Focus Identification) provides several focus categories for disputes and gives an event sentence to predicts the focus category. **MDI** (Marital Disputes Identification) assigns a classification label to the sentence for marital events and predicts the sentence category. **ED** (Event Detection) provides several event types and makes the model determine which event type is involved. The evaluation metric is F1-Score, shown in Formulas 13, 14, and 15.

**ITI** (Issue Topic Identification) provides a set of consultation category labels and legal questions, then predicts the consultation category of the given sentence. The evaluation metric is Accuracy, shown in Formula 20.

The 2nd group focuses on the model's performance on other types of tasks in the full legal domain, including 2 extraction tasks, RC and NER, 2 generation tasks, OS and Co, and 1 regression task, CDC. Results are shown in Table 7. The evaluation details and metrics for each task are as follows:

**RC** (Reading Comprehension) gives an event and a related question, and the task is to answer the question. Most of the questions are focused on information extraction. The evaluation metric is rc-F1, as shown in Formulas 13, 16 and 17.

**NER** (Named-Entity Recognition) focuses on identifying multiple named entities, such as criminals, victims, stolen currency, time, and location. The evaluation metric is soft-F1, as shown in Formulas 13, 18 and 19.

**OS** (Opinion Summarization) extracts key elements from input to generate event summaries. **Co** (Consultation) involves answering questions and providing reasons. The evaluation metric is ROUGE-L, as shown in Formulas 21, 22 and 23.

**CDC** (Criminal Damages Calculation) involves summing the criminal financial amount in the examples to evaluate the model's numbers extraction and mathematical ability. The evaluation metric is Accuracy, as shown in Formula 20.

**Baselines.** We selected six models for performance comparison, including the base model DeepSeek-R1-Distill-Qwen-7B (DeepSeek-7B) and five legal models of similar parameter scale:

Table 6: Comparison of Single and Multiple Task Training under Different Tasks.

| Model | Criminal | | Full Domain | | | | | |
|---|---|---|---|---|---|---|---|---|
| | FAP | CP | DFI | MDI | ITI | RC | OS | Co |
| DeepSeek-7B | 16.86 | 28.89 | **27.20** | 39.69 | 35.80 | **50.83** | 31.66 | 15.10 |
| $M_2$-single | 0.00 | **59.03** | 0.00 | 3.60 | 0.00 | 0.00 | 0.01 | 0.05 |
| $M_2$-multi | **88.94** | 57.05 | 20.20 | **46.24** | **40.20** | 42.14 | **43.10** | 15.29 |

Table 7: Scores for criminal law and full domain law tasks on Lawbench.

| Model | Criminal | | Full Domain | | | | |
|---|---|---|---|---|---|---|---|
| | FAP | CP | RC | NER | OS | CDC | Co |
| DeepSeek-7B | 16.86 | 28.89 | 50.83 | **63.90** | 31.66 | **76.60** | 15.10 |
| LawGPT-beta1.1-7B | 0.15 | 15.68 | 2.27 | 2.00 | 8.61 | 15.40 | 7.62 |
| LexiLaw-6B | 13.15 | 39.99 | 45.39 | 48.74 | 33.12 | 35.80 | 15.82 |
| Fuzi-Mingcha-7B | 25.19 | 55.93 | **97.59** | 44.07 | **54.32** | 47.20 | **16.64** |
| AA-LeLLM-7B | **88.94** | **57.05** | 42.14 | 14.10 | 43.10 | 32.80 | 15.29 |

LaWGPT-7B-beta1.1, LexiLaw, HanFei, Wisdom-Interrogatory and Fuzi-Mingcha. In the Law-Bench paper, the Fuzi-Mingcha model was the best-performing 7B legal domain model in their evaluation. As our research progressed, we also conducted supplementary tests on newly emerged SOTA models, such as Qwen3-8B and Internlm3-Instruct-8B.

After checking the output, we found that the DeepSeek model outputs the $< think >$ tag along with the regular output, affecting evaluations. Therefore, we designed a script to process the output, removing the content within the $< think >< think/ >$ tag to only maintain the valid output, which ensure the normal operation of the evaluations.

**Datasets.** The metrics that we used in benchmarks use CAIL2018, CAIL2019, CAIL2021, CAIL2022, LAIC2021, LEVEN, CrimeKgAssitant, AIStudio, hualv.com. And we use DISC-Law-SFT, fuziminghca to train our model, use lawlib in our CCs-RAG.

### A.3.2 EVALUATION METRICS

**F1-score** FAP (Fact-based Article Prediction), CP (Charge Prediction), DFI (Dispute Focus Identification), MDI (Marital Disputes Identification) and ED (Event Detection) use this metric, as shown in Formulas 13, 14, and 15. $\epsilon$ is a small constant (e.g., $10^{-10}$) to prevent division by zero. TP is true positive. FP is false positive. FN is false negative.

$$F_1 = \frac{2 \times \text{Precision} \times \text{Recall}}{\text{Precision} + \text{Recall} + \epsilon} \quad (13)$$

$$\text{Precision} = \frac{\text{TP}}{\text{TP} + \text{FP}} \quad (14)$$

$$\text{Recall} = \frac{\text{TP}}{\text{TP} + \text{FN}} \quad (15)$$

**rc-F1** RC (Reading Comprehension) use this metric, as shown in Formulas 13, 16 and 17. $|P|$ is the total number of tokens in the predicted answer. $|R|$ is the total number of tokens in the reference(answer). $|S|$ is the number of overlapping tokens between the predicted answer and the reference.

$$\text{Precision} = \frac{|S|}{|P|} \quad (16)$$

$$\text{Recall} = \frac{|S|}{|R|} \quad (17)$$

**soft-F1** ER (Named-Entity Recognition) use this metric, as shown in Formulas 13, 18 and 19. $P$ represents the set of predicted legal entities by the model. $R$ represents the set of actual legal entities in the reference (answer).

$$\text{Precision} = \frac{\sum_{i \in P \cap R} F1_i}{|P|} \tag{18}$$

$$\text{Recall} = \frac{\sum_{i \in P \cap R} F1_i}{|R|} \tag{19}$$

**Accuracy** ITI (Issue Topic Identification) use this metric, as shown in Formula 20. TP is true positive. FP is false positive. FN is false negative. FN is false negative.

$$\text{Accuracy} = \frac{\text{TP} + \text{TN}}{\text{TP} + \text{FP} + \text{FN} + \text{TN}} \tag{20}$$

**ROUGE-L** OS (Opinion Summarization), Co (Consultation) use this metric, as shown in Formulas 13, 18 and 19. LCS (Longest Common Subsequence) is a sequence that appears in the same relative order in both input sequences, but not necessarily consecutively. We choose $\beta = 1$ because there is no specific need to favor either Precision or Recall over the other.

$$F_{\text{lcs}} = \frac{(1 + \beta^2) \cdot R_{\text{lcs}} \cdot P_{\text{lcs}}}{R_{\text{lcs}} + \beta^2 \cdot P_{\text{lcs}}} \tag{21}$$

$$P_{\text{lcs}} = \frac{|LCS(P, R)|}{|P|} \tag{22}$$

$$R_{\text{lcs}} = \frac{|LCS(P, R)|}{|R|} \tag{23}$$

## A.4 MORE DETAILS IN EXECUTION PROCESS

### A.4.1 DATA COLLECTION AND DATA PREPROCESSING

The data for the Incremental Pretraining (IPT) consists of 57k full domain Chinese legal articles, 452 articles of the criminal law and 1k legal case documents. The data is processed into a JSON format containing only the "text" attribute, which is suitable for model pretraining.

Considering that these model pretraining data is insufficient, may resulting in poor performance. We have used a web scraper to collect legal case documents from the website, collecting 172k documents.

In the Supervised Fine-Tuning (SFT), we applied part of the CAIL2018 training datasets and DISC training dataset. The data was divided into nine parts. The first five tasks used CAIL, where key information in the documents was masked to predict the charges, relevant legal article numbers, criminals, fines, and sentencing. The remaining four tasks were processed using DISC, generated 13k criminal law QA events analyses, 19k full legal domain QA events analyses, and 66k and 139k full legal domain QA tasks.

Although we only trained the model with the training set, considering that DISC and 2 evaluation tasks use the same dataset. To ensure the final accuracy, we converted the text into TF-IDF (Term Frequency-Inverse Document Frequency) vectors and computed the cosine similarities

$$\cos(\vec{\alpha}, \vec{\beta}) = \frac{\vec{\alpha} \cdot \vec{\beta}}{\|\vec{\alpha}\| \|\vec{\beta}\|} \tag{24}$$

for the overlapping portions of the training set and the test set, with a threshold set at 0.5. We find that no segments in the test sets exceeded this threshold.

Table 5 provides detailed descriptions of each dataset used in our project and specifies which data were utilized for training each weight of the model.

### A.4.2 TRAINING PROCESS

DeepSeek has demonstrated strong general-domain capabilities and shows promising performance on legal-domain tasks even before fine-tuning, as shown in Table 5. To further enhance its effectiveness on our AALawyer-related legal tasks, we incremental pretrained and finetuned it using domain-specific legal datasets via LlamaFactory (Zheng et al., 2024),

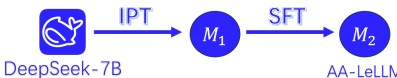

Figure 7: Model Training Flowchart

enabling the model to better adapt to our target objectives. Followed by the normal pipeline described in Section 2.2, our entire process is shown in Figure 7.

**Stage 1 Incremental Pretraining (IPT).** Our goal was to adapt the base DeepSeek-R1-Distill-Qwen-7B model (Guo et al., 2025) to the legal domain's language and style. We used the datasets described in Section A.4.1 for unsupervised Causal Language Modeling (CLM), with the training objective

$$L_{\text{IP}} = -\sum_{t=1}^{T} \log P_\theta(y_t \mid y_{<t}). \tag{25}$$

This process, which resulted in model $M_1$, was conducted using LoRA (Hu et al., 2022).

For the LoRA adaptation, we set the rank $r$ to 8, the scaling factor $\alpha$ to 16, and the dropout rate to 0. We used the AdamW optimizer with a learning rate of $5 \times 10^{-5}$ and a cosine learning rate scheduler. The training was conducted with a per-device batch size of 2, 8 gradient accumulation steps, and a sequence length of 2048.

**Stage 2 Supervised Finetuning (SFT).** We then finetuned model $M_1$ to create our final model, AA-LeLLM, improving its performance on the legal tasks we are focusing on. Recognizing that single-task training was ineffective for other tasks, we employed multi-task training (Caruana, 1997; Yue et al., 2023) by mixing all SFT datasets. The SFT and multi-task learning objectives are

$$L_{\text{SFT}} = -\sum_{t=1}^{T} \log P_\theta(y_t \mid x, y_{<t}) \tag{26}$$

and

$$L_{\text{MTL}} = \sum_{i=1}^{N} \lambda_i \cdot L_{\text{SFT}}^{(i)}. \tag{27}$$

The per-device batch size was set to 1, while other settings remained consistent with the IPT stage.

### A.5 LIMITATIONS AND FUTURE WORK

Due to limitations in computational resources and datasets, the current model has only been experimented on criminal law data using a 7B-scale model. However, it can be extended to other legal fields or even other application domains.

One limitation is that our training data is focused on classification and analysis tasks within criminal law to verify our RAG approaches. While this specialization leads to strong performance on our target tasks, it restricts the model's generalization to other legal domains and tasks. Future work will involve expanding and adjusting our dataset to enrich the training materials, cover more legal domains tasks, annotating data that aligns with our model's analysis structure to enhance post-RAG effectiveness.

For our proposed HR-Benchmark, we use the LLM-as-a-Judge approach. While this method allows for professional and consistent evaluation, addressing our experts limitation. We acknowledge it has limitations. There has the potential for bias and the risk of circular validation. If possible, further work should incorporate a small-scale evaluation evaluated by legal experts to cross-verify the results from our LLM-as-a-Judge approach.

Furthermore, a key area for future work is achieving a truly training-free deployment for AC-RAG. Our ablations show that with the development of LLM, this goal is becoming more achievable. We

can make the fine-tuning process cost less, or even find a proper prompt or another collaborative method to complete training-free deployment.

## A.6 CLARIFY OF LLMS USAGE

We used ChatGPT and Gemini to improve the phrasing and grammar of our paper during the writing process. In the experimental phase, these models were also used to debug and modify our code. All content is authentic, and the final wording and the effectiveness of the code and results have been checked by us.

Table 8: Output of AALawyer

| |
|---|
| **Input(Case):** Concert tickets turn out to be a blank cheque "Scalpers" buy concert tickets on behalf of the "mystery". 
 In October 2023, the defendant Gao Moqi published a false advertisement for the sale of concert tickets of a star's fans in the wechat group, and the victim Xiao Ji (a pseudonym) contacted him to buy concert tickets after seeing it. Gao Moqi defrauded Xiao Ji of a total of 4,500 yuan by issuing tickets. In November of the same year, the defendant Gao Mouqi was seized by the public security organs and was released on bail on the same day. In July 2024, the defendant Gao Mou Qi resumed his old business, published false advertisements for the sale of concert tickets under a star's microblog dynamics, and defrauded multiple victims of a total of 19,723 yuan by issuing tickets and unfreezing accounts. |
| **Related Articles:** 
 Article 266 Whoever defrauds public or private property, if the amount involved is relatively large, shall be sentenced to fixed-term imprisonment of not more than three years, criminal detention or public surveillance and shall also, or shall only, be fined; If the amount involved is huge, or if there are other serious circumstances, he shall be sentenced to fixed-term imprisonment of not less than three years but not more than 10 years and shall also be fined; If the amount involved is especially huge or if there are other especially serious circumstances, he shall be sentenced to fixed-term imprisonment of not less than 10 years or life imprisonment and shall also be fined or have property confiscated. Where this Law provides otherwise, such provisions shall prevail. |
| **Case Analysis:** 
 The defendant Gao Mouqi is guilty of fraud. According to Article 266 of the Criminal Law, whoever, with the purpose of illegal possession, defrauds another person of money or property, if the amount involved is relatively large, shall be sentenced to fixed-term imprisonment of not more than three years, criminal detention or public surveillance and shall also, or shall only, be fined. Gao Mou Qi defrauded Xiao Ji with a total of 4,500 yuan on the grounds of ticket issuance, and defrauded a number of victims with a total of 19,723 yuan on the grounds of ticket issuance and unfreezing accounts, which is a large amount, which is in line with the constitutive requirements of the crime of fraud. Therefore, Gao should be sentenced to less than three years in prison and fined. |
| **Similar Cases:** 
 (Too long, shown in Table 9) |

Table 9: Similar Cases

(2020) Shanghai No. 0101 Criminal Case 621

The public prosecutor is the People's Procuratorate of Huangpu District, Shanghai.

Defendant Wang 2, female, born on March 9, 1997, of the Yi ethnic group, has her registered residence in Yunnan Province.

The defense attorney is Sun Peng, a lawyer from Shanghai Hushi Law Firm.

The People's Procuratorate of Huangpu District, Shanghai Municipality, with the indictment No. 80 of criminal prosecution of Huangpu District, Shanghai Municipality [2020], charged the defendant Wang 2 with fraud and filed a public prosecution with this court on September 3, 2020. This court lawfully applied the simplified procedure, conducted a trial by a single judge, and held a public hearing on this case. The People's Procuratorate of Huangpu District, Shanghai Municipality, assigned prosecutor Chen Mou 2 to support the public prosecution in court. The defendant Wang 2 and his defense lawyer Sun Peng attended the trial. The case has now been concluded.

The People's Procuratorate of Huangpu District, Shanghai Municipality, has charged that between June 2018 and September 2019, the defendant Wang 2 repeatedly posted false information online claiming to have concert tickets for sale and fabricated a counterfeit "Zhuanzhuan" second-hand trading website page to deceive multiple victims who were seeking to purchase tickets. Subsequently, Wang 2 sent false payment links from the "Zhuanzhuan" website to the victims, defrauding them of a total of 24,125 yuan (the currency used hereinafter is the same). After obtaining the money, Wang 2 cut off contact with the victims and squandered all the ill-gotten gains. The specific facts are as follows:

1. On June 14, 2018, the defendant Wang 2 falsely claimed on Weibo that he had EXO concert tickets for sale, defrauding the victim Zhong Moumou of 4,022 yuan for ticket purchases. ...... On September 30, 2019, the defendant Wang 2 fabricated on Weibo that he had tickets for Troye Sivan's concert in Chengdu and defrauded the victim Zhou 3 of 1,498 yuan for the ticket purchase. On April 28, 2020, the public security authorities arrested the defendant Wang 2 in Kunming City, Yunnan Province. After being apprehended, with the assistance of his family, Wang 2 has returned all the ill-gotten gains.

The above facts are confirmed by the statements of the victims Zhong Moumou, Wan Moumou, Li Moumou, Xu Moumou, Dai Moumou, Man Moumou, Chen Mou1, Zhang Moumou, Shen Moumou, Zhou Mou1, Xue Mou, Yuan Moumou, Zhou Mou2, and Zhou Mou3; the testimony of witness Wang 1; screenshots of relevant WeChat chat records; screenshots of WeChat transfer records; the seizure decision and list issued by the Cultural Security Division of the Shanghai Municipal Public Security Bureau; the working situation issued by the public security authorities; the seizure decision, list of seized property and documents issued by the People's Procuratorate of Huangpu District, Shanghai; the letter of forgiveness; and the multiple confessions of the defendant Wang 2.

The public prosecutor believes that the defendant Wang 2, with the intent of illegal possession, fabricated facts and concealed the truth to defraud others of their property in a relatively large amount. His actions have violated Article 266 of the Criminal Law of the People's Republic of China and he should be held criminally responsible for fraud. The defendant Wang 2 has confessed to the crime and accepted the punishment. According to Article 15 of the Criminal Procedure Law of the People's Republic of China, he may be given leniency. The defendant Wang 2 has truthfully confessed to the criminal facts. According to Article 67, Paragraph 3 of the Criminal Law of the People's Republic of China, he may be given a lighter punishment. It is suggested that Wang 2 be sentenced to 11 months in prison and fined 2,000 yuan.

The defendant Wang 2 has no objection to the facts, evidence, charges and sentencing suggestions made by the public prosecutor regarding his crime of fraud and has signed and confirmed them. During the court hearing, he also expressed no objection. The defense lawyer has no objection to the facts, evidence, charges and sentencing suggestions made by the public prosecutor. The lawyer believes that the defendant Wang 2 can truthfully confess his criminal facts, has returned all the ill-gotten gains and has obtained the forgiveness of the victims Li Moumou and Zhang Moumou, and thus should be given a lighter punishment.

This court holds that the defendant Wang 2, with the intent of illegal possession, repeatedly posted false information on the internet and concealed the truth, defrauding multiple victims of a considerable amount of property. His actions have violated the criminal law and constitute the crime of fraud, for which he should bear criminal responsibility. The facts charged by the public prosecutor against the defendant Wang 2 for fraud are clear, the evidence is solid and sufficient, and the charge is valid. This court supports it. The defendant Wang 2 can truthfully confess his crime after being arrested, and thus can be given a lighter punishment; he can also accept the punishment and thus can be given leniency. The defense lawyer's opinion that the defendant Wang 2 should be given a lighter punishment is adopted. Therefore, in accordance with Article 266 and Article 67, Paragraph 3 of the Criminal Law of the People's Republic of China,

