# OpenReview forum: "AALawyer: A Generative Retrieval-Augmented Large Language Model System for Legal Reasoning"
_ICLR.cc/2026/Conference — ICLR 2026 Conference Withdrawn Submission_

### Official Review · Reviewer_54pV · 2025-10-20

**Soundness:** 2
**Presentation:** 2
**Contribution:** 2
**Rating:** 4
**Confidence:** 3

**Summary:**

This paper presents AALawyer, a generative retrieval-augmented legal AI system that combines a fine-tuned legal LLM, precise legal and case retrieval modules, and a 4-dimensional evaluation benchmark to improve accuracy, reduce hallucinations, and enhance explainability in criminal law reasoning.

**Strengths:**

1. This paper introduces a new RAG method. The ARTICLE-CONTENT RAG provides insights.
2. The experiment shows the effectiveness of the proposed method.

**Weaknesses:**

1. **Unclear definition of hallucination measurement.**
   In Eq. (3), the calculation of `auth()` is not clearly introduced. Does it originate from the article itself, or is it derived through another computational method? As this function is a crucial component of the AC-RAG module, it should be explained in detail. Providing a concrete example would also help readers better understand the process.

2. **Lack of clarity in presentation.**
   The paper is somewhat difficult to follow. For instance, Figure 1 provides limited information and would be more informative if it used a legal-related context. Similar issues are observed with Figure 2.

3. **Minor inconsistencies and typos.**
   There are inconsistent expressions such as *G&G RAG* and *G&G-RAG*. It is recommended to standardize these terms throughout the paper for consistency and professionalism.

4. **Outdated baselines.**
   Some comparison methods appear outdated. To the best of my knowledge, the experiments do not include the latest Legal LLMs [1], and the LawGPT method has been updated in [2]. It would strengthen the paper to incorporate these recent models in the experimental comparison.

***
Ref:

[1] Chen, H., Xu, Y., Wang, B., Zhao, C., Han, X., Wang, F., ... & Xu, Y. (2025). LexPro-1.0 Technical Report. arXiv preprint arXiv:2503.06949.

[2] Zhou, Z., Yu, K. Y., Tian, S. Y., Yang, X. W., Shi, J. X., Song, P., ... & Li, Y. F. (2025). Lawgpt: Knowledge-guided data generation and its application to legal llm. arXiv preprint arXiv:2502.06572.

**Questions:**

See weaknesses above

---

### Official Review · Reviewer_5u4H · 2025-11-01

**Soundness:** 2
**Presentation:** 2
**Contribution:** 2
**Rating:** 2
**Confidence:** 4

**Summary:**

This paper proposes AALawyer, a legal reasoning system for Chinese criminal law that combines a fine-tuned legal LLM (AA-LeLLM) with two retrieval-augmented generation (RAG) modules: AC-RAG (Article-Content RAG) and CCs-RAG (Case-Cases RAG). The AC-RAG module converts article retrieval into a classification task to reduce hallucination in legal citations, while CCs-RAG retrieves similar precedent cases using a dataset of 176k criminal court cases. The authors evaluate their system on LawBench classification tasks and introduce a new Hallucination Risk-Benchmark (HR-Benchmark) with four dimensions: hallucination score, professionalism, informativeness, and explainability. The results show improvements over baseline models.

**Strengths:**

1. **Well-motivated approach**: The paper grounds its design in legal syllogism theory and real-world judicial practice, providing a principled framework for the three-stage pipeline (retrieving legal articles, referencing precedent cases, and generating conclusions).

2. **Comprehensive system design**: The integration of classification-based article retrieval (AC-RAG) with case-based retrieval (CCs-RAG) addresses both hallucination reduction and explainability, which are critical for legal applications.

3. **Strong empirical results on target tasks**: The model achieves impressive performance on the criminal article prediction task (FAP), with a 71.98% improvement over the base model, demonstrating the effectiveness of the proposed AC-RAG approach.

**Weaknesses:**

1. **Limited novelty in core concept**: The idea of incorporating legal articles into legal reasoning to reduce hallucination is not new. Previous work such as Lawyer LLaMA (Huang et al., 2023) has already explored similar methods of retrieving and integrating legal provisions into the generation process.

2. **Questionable benchmark validity**: The construction and evaluation methodology of the Hallucination Risk-Benchmark is insufficiently rigorous and lacks credibility as a reliable evaluation framework:

* Inadequate dataset construction details: The paper states that 200 cases were "randomly selected" from CAIL2018 but provides no justification for this sampling strategy. Critical questions remain unanswered: What criteria ensure these cases have appropriate difficulty levels? How is diversity guaranteed—do they cover a broad range or representative set of criminal charges? Why only 200 cases rather than a more comprehensive evaluation set? Without this information, the benchmark's representativeness and reliability are questionable.

* Ambiguous and unvalidated evaluation dimensions: The four dimensions (hallucination, professionalism, informativeness, explainability) are vaguely defined without clear operational definitions or grounding in legal theory. It is unclear whether these dimensions are legally meaningful or simply intuitive categories. The paper does not justify why these specific four dimensions are appropriate for evaluating legal reasoning systems.

* Insufficient LLM-as-judge methodology: While the authors employ an LLM-as-judge approach, Section 4 provides no details about the evaluation prompts or scoring rubrics. The validity of using LLMs to evaluate legal reasoning quality—especially for domain-specific dimensions like "professionalism"—is not established. Critically, there is no validation against legal expert judgments to confirm that the LLM's assessments align with human expert evaluations. This is a significant omission given that the benchmark is proposed as a contribution of the paper.

**Questions:**

Please refer to the weaknesses.

---

### Official Review · Reviewer_ghT2 · 2025-11-01

**Soundness:** 2
**Presentation:** 1
**Contribution:** 3
**Rating:** 4
**Confidence:** 3

**Summary:**

Authors present AALawyer, a three-stage legal RAG system: (1) AC-RAG — a classifier that predicts article numbers and retrieves canonical article text, (2) CCs-RAG — case-to-case retrieval, and (3) AA-LeLLM — a finetuned legal generator. They introduce an HR-Benchmark (4 dimensions including a “Hallucination” score) and report improved scores versus some baselines. The central claim is that their generative RAG (AC-RAG + CCs-RAG + AA-LeLLM) reduces hallucination and improves explainability for criminal-law reasoning. I

**Strengths:**

1.Thoughtful pipeline architecture. The tri-stage structure is intuitively aligned with how legal professionals think: identify governing statute, retrieve precedent, and then craft reasoned text. This mapping from professional workflow to system architecture is a strong design.

2. Focus on hallucination risk. The inclusion of HR benchmark and metrics signals good awareness of the major failure mode in legal generative systems. The fact that the authors treat hallucination explicitly is a positive step.

**Weaknesses:**

1. Figure 1(b) can have some additional illustrations. What are R and G? Can you also attach a clearer message on what the difference this approach aims at achieving?
2. The terminologies in this paper are really confusing. There are two module names, and there's also AA-LeLLM, along with other abbreviations like Auth, CP, FAP, ITI, DFI, and the four dimensions of your eval also comes in abbreviations, just leaving the paper with an overwhelming impression. I find it hard to jump back and forth when I read it. The paper will improve presentation a lot by simplifying terminologies.
3. The paper structure is atypical and interrupts the flow of the article. I would not recommend putting related work between your method and experiments.
4. It also seems slightly unfair that the paper is comparing a LM system with off-the-shelf LMs. Should have compared with stronger baselines which are also LM systems. This undermines the validity and meaningfulness of the comparison.

**Questions:**

Suggestions: I really encourage the authors to simplify the formulation of this paper as it feels needlessly complicated. Fonts in Figure 2 are also a bit too small. The improvement in accuracy are also less valid and meaningful since the comparison is a bit unfair between LM systems and off-the-shelf models.

---

### Official Review · Reviewer_6rcM · 2025-11-02

**Soundness:** 3
**Presentation:** 2
**Contribution:** 2
**Rating:** 4
**Confidence:** 4

**Summary:**

This paper addresses hallucinated citations and limited explainability in legal LLMs by proposing AALawyer—a generative retrieval-augmented system for criminal law reasoning (integrating AA-LeLLM, AC-RAG, CCs-RAG)—and HR-Benchmark (4-dimensional evaluation). Experiments show AALawyer achieves 88.84% accuracy on LawBench’s FAP task (↑71.98% vs. DeepSeek-7B) and reduces hallucination risk by 37.6% on HR-Benchmark. Key contributions include the novel AC-RAG pipeline, 176k criminal case dataset, and HR-Benchmark.

**Strengths:**

1. AC-RAG innovates a "generate-retrieve-generate" pipeline, using numeric identifiers to convert article retrieval into a classification task, enabling parameter sharing and reducing hallucination.
2. The 176k case dataset fills gaps in criminal law retrieval data; HR-Benchmark addresses the lack of RAG system evaluation tools in legal AI.
3. Experiments focus on core FAP tasks; ablation studies (e.g., AC-RAG cuts hallucination by 59%) and Lemma 1 (mathematical proof of hallucination bounds) ensure rigor.
4. The 3-stage pipeline (AC-RAG/CCs-RAG/AA-LeLLM) and theoretical foundations are clearly presented.

**Weaknesses:**

1. Generalization Limits: AA-LeLLM is only based on 7B DeepSeek-7B (no 13B/70B analysis) and restricted to criminal law (no discussion on extending to civil/administrative law).
2. HR-Benchmark Reliability: Relies solely on DeepSeek-Chat for evaluation (no scoring criteria or multi-Judge comparison); 200 CAIL2018 cases lack clarity on covering major crimes.
3. AC-RAG Gaps: Fails to quantify numeric identifier prediction errors or disclose Format1/Format2 prompt templates; no analysis of "ambiguous cases".
4. Related Works: Omits discussion of domain association models like Hyper-RAG, which suit legal complex relationships.

**Questions:**

see weaknesses

---

### Note · Authors · 2025-11-13

I have read and agree with the venue's withdrawal policy on behalf of myself and my co-authors.